# *In vivo* synthesis of bacterial amyloid curli contributes to joint inflammation during *S*. Typhimurium infection

**Amanda L. Miller**[1], **J. Alex Pasternak**[2,¤b], **Nicole J. Medeiros**[1,¤c], **Lauren K. Nicastro**[1], **Sarah A. Tursi**[1,¤c], **Elizabeth G. Hansen**[2,3], **Ryan Krochak**[2,3], **Akosiererem S. Sokaribo**[2,3], **Keith D. MacKenzie**[2,3], **Melissa B. Palmer**[2,3], **Dakoda J. Herman**[2,3], **Nikole L. Watson**[2,3], **Yi Zhang**[1], **Heather L. Wilson**[2], **R. Paul Wilson**[1,¤a], **Aaron P. White**[2,3]*, **Çagla Tükel**[1]*

**1** Department of Microbiology and Immunology, Lewis Katz School of Medicine, Temple University, Philadelphia, Pennsylvania, United States of America, **2** Vaccine and Infectious Disease Organization-International Vaccine Centre, Saskatoon, Saskatchewan, Canada, **3** Department of Biochemistry, Microbiology and Immunology, University of Saskatchewan, Saskatoon, Saskatchewan, Canada

☯ These authors contributed equally to this work.
¤a Current address: GlaxoSmithKline, Collegeville, Pennsylvania, United States of America
¤b Current address: Department of Animal Sciences, Purdue University, 270 S. Russell St, West Lafayette, Indiana, United States of America
¤c Current address: Jansson Research & Development, LLC, Spring House, Pennsylvania, United States of America
* aaron.white@usask.ca (APW); ctukel@temple.edu (CT)

**Data Availability Statement:** All relevant data are within the manuscript and its Supporting Information files.

## Abstract

Reactive arthritis, an autoimmune disorder, occurs following gastrointestinal infection with invasive enteric pathogens, such as *Salmonella enterica*. Curli, an extracellular, bacterial amyloid with cross beta-sheet structure can trigger inflammatory responses by stimulating pattern recognition receptors. Here we show that *S*. Typhimurium produces curli amyloids in the cecum and colon of mice after natural oral infection, in both acute and chronic infection models. Production of curli was associated with an increase in anti-dsDNA autoantibodies and joint inflammation in infected mice. The negative impacts on the host appeared to be dependent on invasive systemic exposure of curli to immune cells. We hypothesize that *in vivo* synthesis of curli contributes to known complications of enteric infections and suggest that cross-seeding interactions can occur between pathogen-produced amyloids and amyloidogenic proteins of the host.

## Author summary

Our manuscript focuses on curli, a 'functional amyloid' produced by Salmonella as well as other enteric bacteria. We present the first biochemical evidence that these fibers are produced in the gastrointestinal tract of mice after oral infection, the natural route for Salmonella infections. This finding is significant because of the immune impacts on the host; we show that curli cause an increase in autoimmunity and inflammation in the knee joints of infected mice. Reactive arthritis is a known autoimmune complication after enteric infections and our results indicate that presence of curli in the gut provides a novel linchpin of

**Funding:** This work was supported by NIH grants AI137541, AI125429, AI126133, AI132996, AI148770, AI153325 and AI151893 to CT, by a Discovery Grant from the Natural Sciences and Engineering Research Council of Canada (NSERC; #2017-05737) and the Jarislowsky Chair in Biotechnology to APW. JAP and KDM were supported by fellowships from the Saskatchewan Health Research Foundation. NSERC also provided support to EGH (Undergraduate Research Award) and MBP (Integrated Training Program in Infectious Diseases). RK, DJH, and NLW were each supported by Biomedical Research Awards from the University of Saskatchewan. The funders had no role in study design, data collection and interpretation, or the decision to submit the work for publication. Any opinions, findings, and conclusions or recommendations expressed in this material are those of the author(s) and do not necessarily reflect the views of the NIH or NSERC.

**Competing interests:** The authors have declared that no competing interests exist.

pathogenesis. As curli or curli-like amyloids are also produced by the commensal bacteria, it is possible that the unintended release of amyloids produced by the microbiota could trigger similar autoimmune reactions. Finally, our work provides conceptual evidence for the possibility of cross-seeding between bacterial amyloids like curli and human amyloids involved in amyloid-associated diseases such as Alzheimer's Disease via the gut microbiome or infections.

## Introduction

*Salmonella enterica* serovar (*S.*) Typhimurium is a common non-typhoidal *Salmonella* species (NTS), which causes gastroenteritis (diarrhea) in immunocompetent individuals [1]. Approximately 5% of patients develop an autoimmune condition known as reactive arthritis following gastrointestinal infection with *S.* Typhimurium and some patients remain symptomatic for 5 years or longer [2–6]. *S.* Typhimurium also successfully colonizes multiple surfaces in the environment and cycles between the host and the environment. During the environmental phase of its life cycle, *S.* Typhimurium forms multicellular communities called biofilms; these structures involve a thick extracellular matrix composed of curli, cellulose, BapA, and extracellular DNA (eDNA) that has been shown to protect bacteria from various environmental insults including chemicals, osmotic stress, and oxidative stress [7]. The ability of NTS to form biofilms is predicted to be a conserved strategy for increased persistence and survival in non-host environments, which would increase the likelihood of transmission to a new host [8, 9]. Previous work proposed that biofilm formation and aggregation could also provide *Salmonella* with a mechanism to survive the harsh conditions of the host intestinal tract [10–12]. However, whether *Salmonella* forms biofilms or biofilm-like aggregates within the human host is not known.

Curli are highly aggregated, thin amyloid fibers expressed on the surface of enteric bacterial cells [13–16]. These fibers range from 4 to 10 nm in width and have a rich β-sheet structure in which the β-sheet strands are orientated perpendicular to the axis of the fiber [17]. This cross-beta structure is characteristic of amyloids, including the majority of proteins associated with human amyloid diseases [18, 19]. Curli biosynthesis is controlled by a Type VIII secretion system encoded by two divergently transcribed operons, *csgBAC* and *csgDEFG* [7, 20–22]. In this system, CsgD transcriptionally activates both *csgBAC* and *csgDEFG* operons as well as several additional genes important for biofilm formation, such as those involved in cellulose biosynthesis [11, 23–25]. The *csgA* gene encodes the major subunit of curli, CsgA, which forms the curli fibers [26, 27]. The *csgB* gene encodes CsgB, which is the curli nucleator, forming a base for CsgA to polymerize at the cell surface [28]. CsgC is a periplasmic protein that inhibits amyloid formation in the cytoplasm by inhibiting the beta-sheet transition [29]. Once produced, curli forms a mesh-like matrix on the biofilm along with extracellular DNA and cellulose [16, 30, 31].

Routine curli production by *Salmonella in vitro* occurs at temperatures lower than 30˚C [32, 33], suggesting that fiber synthesis is limited to environmental conditions. In contrast, seroconversion studies in mouse models indicate that curli can be expressed during infection [34]. In support of this hypothesis, some *in vitro* conditions can bypass temperature-dependent induction, resulting in curli production at 37˚C when bacteria are grown in iron-limiting conditions [14], in the presence of bile [35], or when mutations exist in the *csgD* promoter [14, 32]. When mice are injected intraperitoneally with purified curli or *S.* Typhimurium that were induced to express curli, mice develop an autoimmune response characterized by anti-double

stranded DNA and anti-chromatin autoantibodies [36]. This response is driven by the innate immune recognition of curli by the two Toll-like receptors, TLR2 and TLR9 [7, 37–40]. In this study, we show that *S.* Typhimurium produces *de novo* curli in the murine gastrointestinal (GI) tract following oral infection and that expression of curli leads to elevated levels of auto-antibodies and inflammation in joints, which are hallmarks of human reactive arthritis. The negative impacts of curli appear to be correlated with cellular invasion and access of curli to the systemic tissues surrounding the intestine.

## Results

### Detection of curli in the intestinal tract of mice infected with *S.* Typhimurium

For many human bacterial pathogens, transmission depends on cycling through the environment. We speculated that preparation for this stage of the lifecycle could be a central strategy for *Salmonella* to ensure efficient and prolonged success as pathogens. This led us to investigate whether *S.* Typhimurium could synthesize curli *in vivo*. Streptomycin pre-treated C57BL/6 mice were orally infected with *S.* Typhimurium grown so that curli were not present in the inoculum. At four to six days after inoculation, mice were successfully infected, as measured by colony counts from the internal organs and feces (S1 Fig). Tissue embedding and sectioning were performed and small bacterial clusters or microcolonies were visible within the cecum (S2 Fig). Multi-color immunohistochemistry staining revealed that these rod-shaped bacterial cells were *Salmonella* and were producing curli (Fig 1A and 1B).

   *Salmonella* were found exclusively within the luminal compartment of the cecum as defined by ZO-1 staining on the apical surface of the epithelium (Fig 1B). Control experiments with non-immune rabbit serum (Fig 1C and 1D) did not show any unspecific staining. We analyzed tissues from additional mice to monitor the progression of *S.* Typhimurium through the GI tract. Dense concentrations of *S.* Typhimurium showing positive staining for curli were detected in the cecum (Fig 1F and 1I). In the colon, the *S.* Typhimurium were detected more tightly packed, had the most intense curli staining, and appeared to be in the lumen (Fig 1G and 1J). In contrast, *S.* Typhimurium were negative for curli production in the small intestine and were sporadically spread out over the 30 cm length of intestine. *S.* Typhimurium were detected in the liver but were not recognized by anti-curli immune serum (Fig 1E and 1H). Overall, these results suggested that *S.* Typhimurium produced curli in the cecum and colon within four to six days after oral infection. However, as these experiments utilized polyclonal antibodies and also lacked control tissues from mice infected with the curli mutant, these observations needed additional confirmation.

### Curli are produced by S. Typhimurium after oral infection of antibiotic-treated or untreated mice

There are two acute models of *Salmonella* infection that can be employed using susceptible (i.e., NRAMP-negative) C57BL/6 mice. With streptomycin pre-treatment of the mice, the infection model pathology more closely approximates human gastroenteritis, including transient disruption of the intestinal microbiota, epithelial ulceration and edema, replication of *S.* Typhimurium to high bacterial densities in the large intestine and high levels of bacterial shedding in the feces [41–44]. In antibiotic untreated C57BL/6 mice, disease progression more closely approximates human typhoid fever, with less intestinal inflammation, coupled with *S.* Typhimurium replication to lower cell densities in the GI tract [45]. We wanted to determine if the host environment in each infection model provided the activating conditions for curli

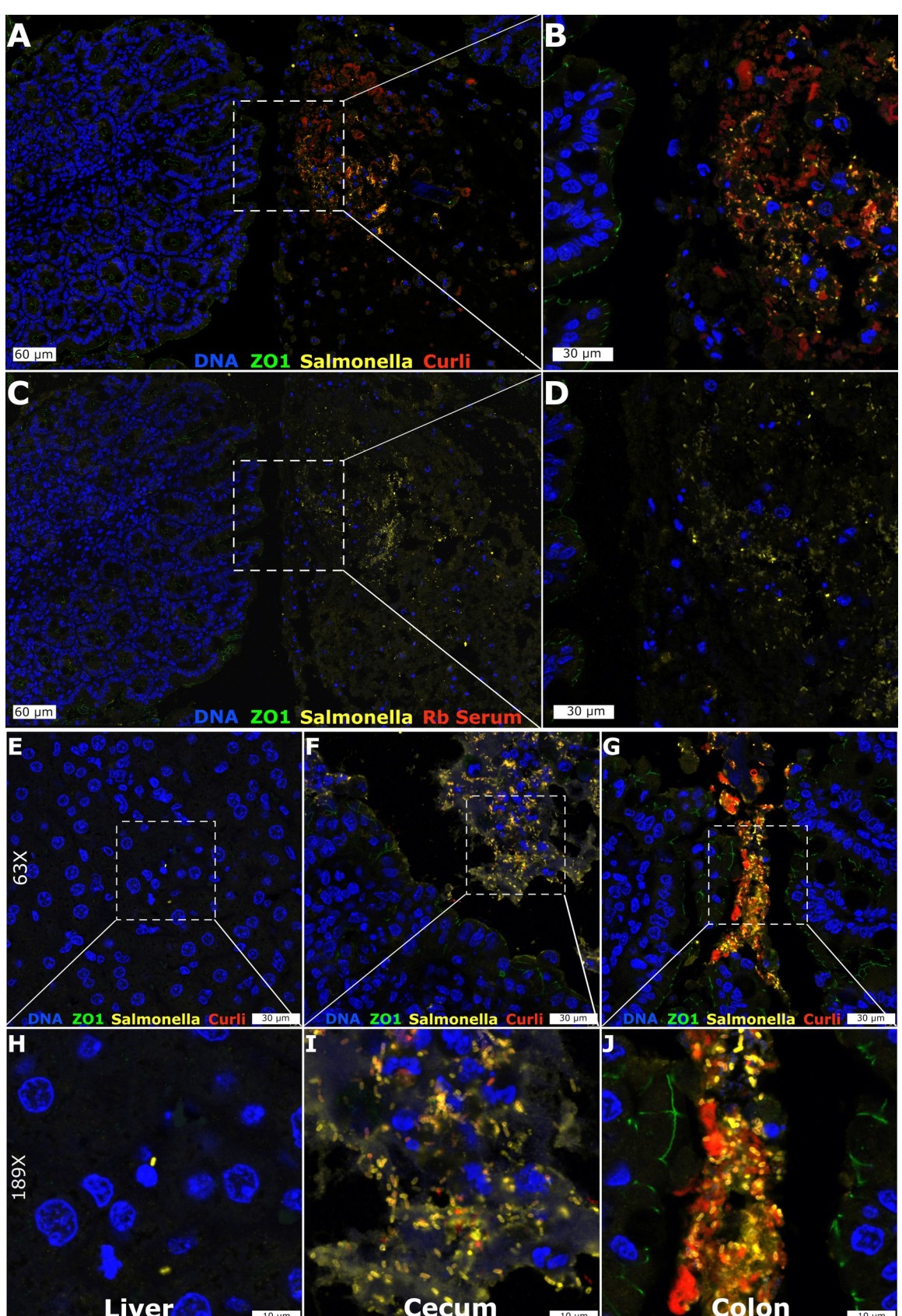

**Fig 1. Immunofluorescent detection of curli produced by *S*. Typhimurium in the mouse gut.** Representative confocal images at 28x (A, C), 63x (E-G), 72x (B, D) and 189x (H-J) are shown for the *S*. Typhimurium-infected mouse Cecum (A-D, F, I), Liver (E, H), and Colon (G, J). Multicolor immunofluorescent staining was directed at ZO-1 (green), Salmonella (yellow) and Curli (red) along with DAPI counterstain (blue). Naïve rabbit serum was used as a control for non-specific staining (C, D).

production. We monitored expression of *csgD*, encoding the curli transcriptional activator [14, 46], using a luciferase reporter strain grown in curli-inducing or non-inducing conditions prior to oral infection of streptomycin pre-treated or untreated C57BL/6 mice. At 96 h post-infection, the GI tract, spleen, and liver were removed and imaged; light production was detected in the GI tracts of all infected mice, but not in the liver and spleen (Fig 2).

This indicated that *S*. Typhimurium expressed *csgD* in the GI tract during the course of infection, irrespective of how bacteria were grown prior to infection or whether mice were pre-treated with antibiotic. It is important to note that sufficiently large numbers of luciferase positive bacteria are needed to detect the light production. Therefore, *S*. Typhimurium may also be expressing the *csgD* gene at systemic organs, but at relatively low density that cannot be detected with this technique.

Curli amyloid fibrils produced by *Salmonella* require treatment with 90% formic acid to depolymerize [32, 47]. We used this unique biochemical property to screen mice for direct evidence that curli were produced *in vivo*. Streptomycin pre-treated and untreated C57BL/6 mice were colonized by *S*. Typhimurium to similar levels post-infection, as measured in the mesenteric lymph nodes, spleen, and liver (Fig 3A).

Consistent with known differences between infection models, the strep-treated mice were shedding significantly higher levels of *S*. Typhimurium into the feces compared to the untreated mice (Fig 3B). Tissues from infected mice were snap frozen, ground into a powder, and the soluble proteins were extracted by boiling in SDS-PAGE sample buffer. The remaining insoluble debris was treated once with 90% formic acid (FA) prior to SDS-PAGE and immunoblotting. In initial tests, a high molecular weight band was detected in the colon samples, corresponding to the top of the SDS-PAGE well (S3 Fig). This was consistent with intact curli fibers that had not depolymerized [32]. In subsequent tests, we performed three successive FA treatments of the insoluble portion of the ground tissue samples, resulting in the detection of dimer and monomer CsgA species in the cecum and colon from both strep-treated and untreated mice (Fig 3C). In total, curli-specific bands were detected in 5 of 6 strep-treated mice and 2 of 6 untreated mice. Curli bands were not detected in the liver or small intestine samples (Fig 3C), which matched the immunohistochemistry data. The most intense CsgA bands were detected from the cecum in the majority of mice, as compared to the colon, but this result appeared to be variable. These experiments confirmed that *S*. Typhimurium produced curli amyloids in the lower GI tract of infected mice.

To rule out the possibility that curli were being produced by a bacterial species other than *Salmonella* that might be present in the microbiota, we screened colon samples from eight control mice that were not colonized by *Salmonella*. All samples were negative for curli (S4 Fig).

## Oral infection with wild type *S*. Typhimurium leads to the generation of anti-dsDNA autoantibodies

We recently demonstrated that systemic exposure to purified curli/DNA complexes or to curli-expressing *S*. Typhimurium triggers autoimmunity and leads to the generation of anti-dsDNA autoantibodies [36, 37]. Since oral ingestion is the natural route of *Salmonella* exposure, we wanted to determine if curli synthesis in a natural infection would also generate an autoimmune response. To allow sufficient time for antibody generation (i.e., >2 weeks),

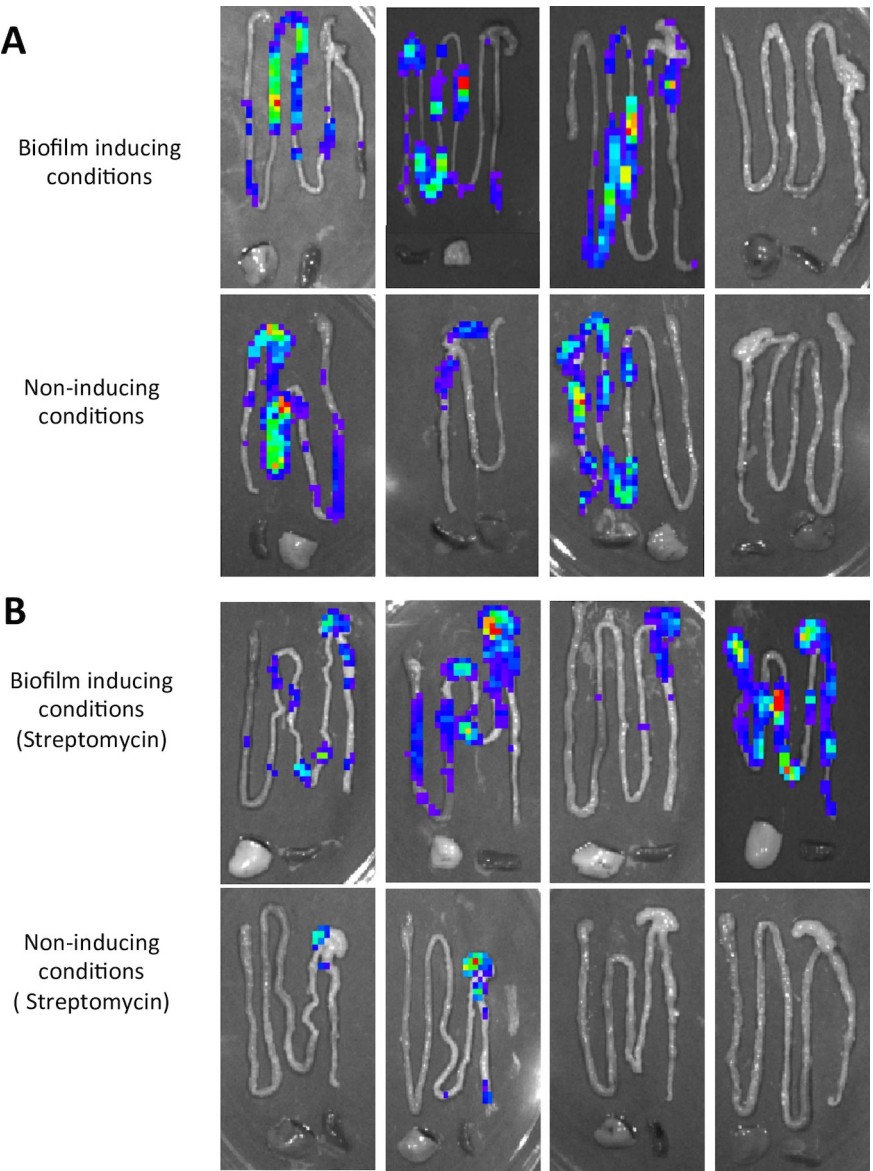

**Fig 2.** *In vivo* **expression of** *csgD* **gene by** *S.* **Typhimurium in the mouse intestinal tract.** Untreated groups of C57BL/6 mice (A) or groups of C57BL/6 mice pre-treated with streptomycin (B) were inoculated orally with $10^8$ CFU of a *S.* Typhimurium *csgD::luxCDABE* reporter strain grown in either biofilm-inducing or biofilm non-inducing conditions. Animals were euthanized at 96 h post-infection. *csgD* expression in the intestinal tract, spleen and liver was measured as light production using an IVIS Spectrum Imaging System (Perkin Elmer). Each panel represents a different mouse.

genetically resistant (i.e., NRAMP-positive) CBA/J mice were orally infected with wild type *S.* Typhimurium or its isogenic curli mutant (ΔcsgBA). Consistent with previous reports [9, 48], bacterial numbers in fecal samples were similar between the wild type and the *csgBA* mutant, with no statistical differences detected for 7 weeks (Fig 4A).

There was also no significant difference in bacterial burdens in spleen and livers from the infected mice after 7 weeks (Fig 4B). To determine if a humoral response was generated against curli during the course of infection, serum from the infected mice at 7 weeks post-infection was used to screen immunoblots containing a CsgA fusion protein (GST-CsgAR1-5; 33 kDa;

[40]). A band was observed in nine out of ten mice infected with wild type *S.* Typhimurium, whereas none of the mice infected with the *csgBA* mutant (*n* = 5) developed anti-curli antibodies (Fig 4C). Wild type-infected mice were positive for CsgA-specific IgG antibodies; as the serum was diluted, the level of IgG antibodies detected was also decreased (Fig 4D). Finally, we

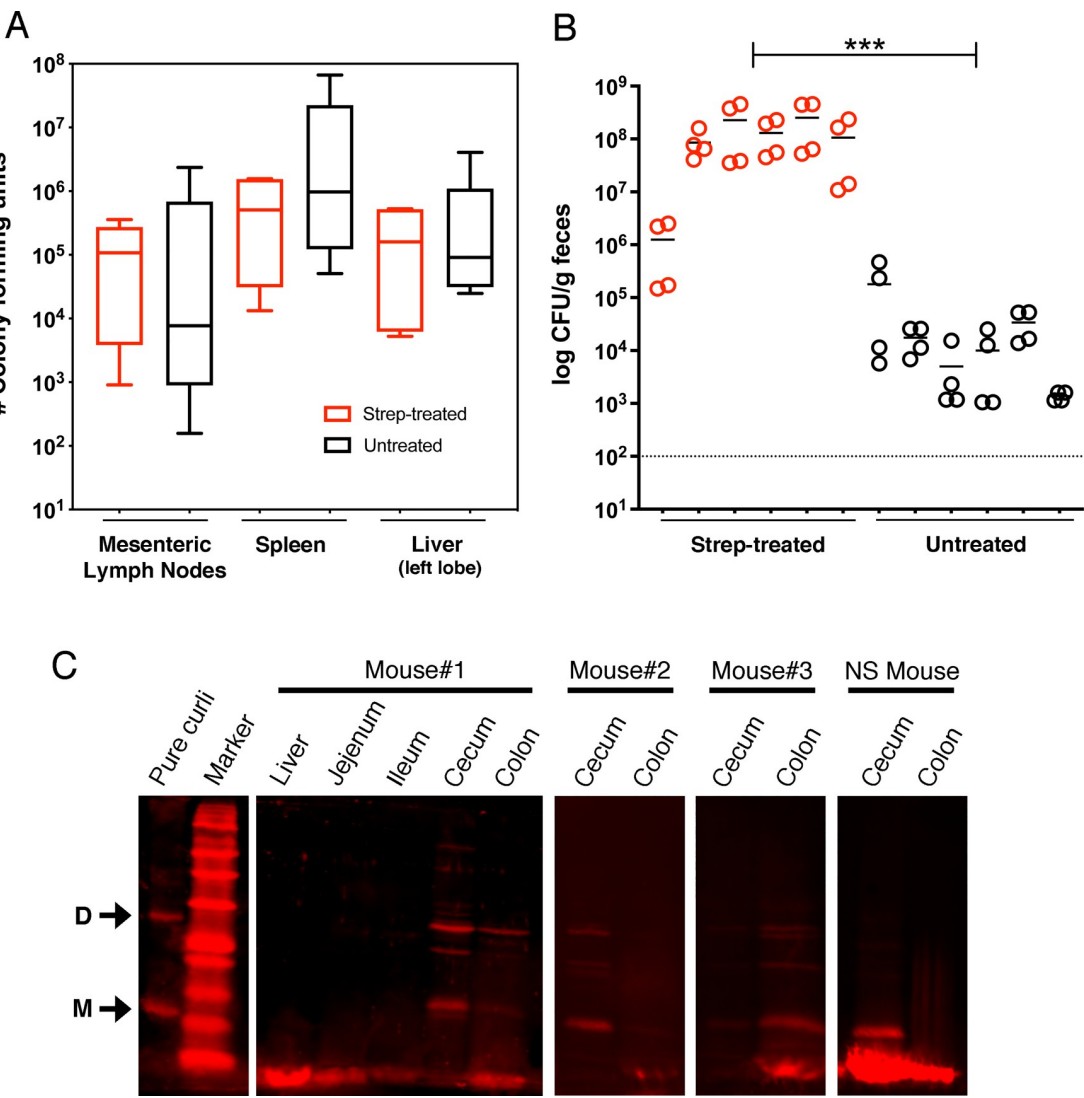

**Fig 3. *In vivo* synthesis of curli by *S.* Typhimurium during infection of C57BL/6 mice.** (A) Groups of C57BL/6 mice (*n* = 6) were orally infected with $10^7$ CFU of *S.* Typhimurium after pre-treatment with streptomycin (red) or untreated (black). Organs were collected and bacteria enumerated from each mouse on Days 4–7 post-infection. The edges of the boxes represent the minimum and maximum CFU values from individual mice within each group, with the line representing the mean and the error bars representing the 95% confidence interval. There was no significant difference between groups for MLN (*P* = 0.3458), spleen (*P* = 0.4125) or liver (*P* = .3997). (B) Fecal pellets were collected on Days 3 and 4 post-infection and the levels of *S.* Typhimurium were determined. Circles represent individual CFU/g measurements with the line representing the mean value from each mouse. The CFU/g values from mice in each group were pooled and compared (*P* = 0.001; \*\*\*). (C) Homogenized mouse tissues prepared for curli analysis were resolved by SDS-PAGE and immunoblotting. Arrows denote the CsgA monomer "M" and dimer "D" species detected in the pure curli control and mouse tissues (i.e., Mouse #1,2,3—Strep-treated; NS mouse–Non-strep-treated). Curli-specific rabbit immune serum was used as primary antibody, followed by IRDye 680RD goat anti-rabbit immunoglobulin G (IgG) secondary antibody and visualization using the Odyssey CLx imaging system and Image Studio 4.0 software package (Li-Cor Biosciences). Representative images are shown.

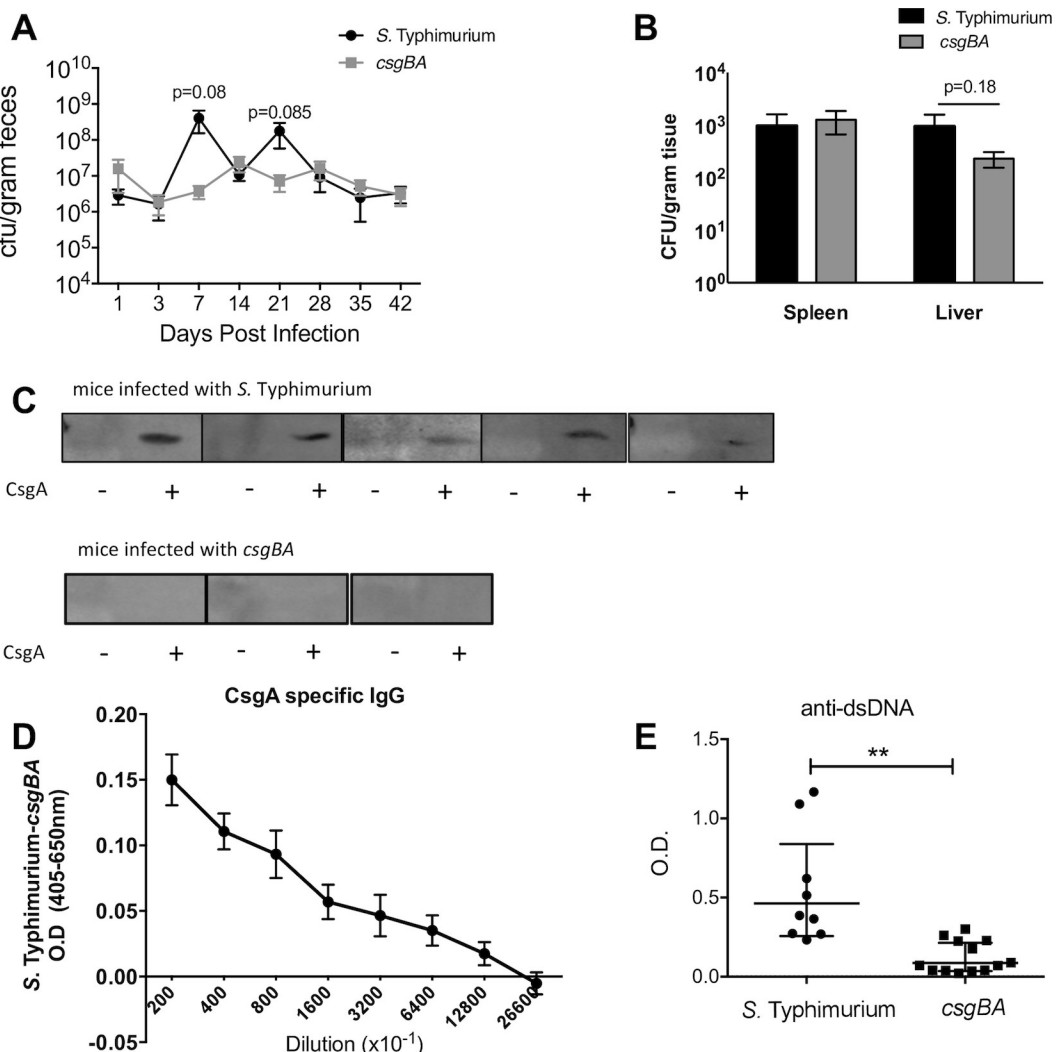

**Fig 4. Long-term persistent infection with *S.* Typhimurium leads to expression of curli and the generation of anti-dsDNA autoantibodies.** Groups of CBA/J mice were orally infected with $10^8$ CFU of *S.* Typhimurium or its isogenic curli mutant (*csgBA*). (A) Fecal pellets were collected on the days shown after infection and the levels of *S.* Typhimurium were determined. (B) Spleen and liver were collected at the experiment end-point (42 days) and bacteria were enumerated. Mean and SE were calculated using data from three independent experiments. (C) SDS-PAGE gels were loaded with a GST (first lane) or GST-CsgAR1-5 fusion protein (second lane). Sera from mice infected with *S.* Typhimurium or it isogenic curli mutant (*csgBA*) for 42 days were used as primary antibodies to detect CsgA production. (D) ELISA was used to quantify the levels of IgG antibodies specific for CsgA in 42-day old serum from mice infected with *S.* Typhimurium (*n* = 13) or the isogenic curli mutant (*csgBA. n* = 10). The absorbance values shown represent the wild type samples after subtraction of the values from the csgBA samples. Mean and SE were calculated using data from three independent experiments. (E) The levels of anti-dsDNA autoantibodies were quantified by ELISA. Mean and SE were calculated using data from two independent experiments. Significance was calculated using a Student's t-test. $^*p < 0.05$, $^{**}p < 0.01$, $^{***}p < 0.001$.

performed an anti-dsDNA ELISA, which showed that mice infected with wild type, curli-producing *S.* Typhimurium developed significantly higher levels of anti-dsDNA autoantibodies compared to the mice infected with the isogenic *csgBA* mutant strain (Fig 4E).

## Curli expression leads to autoimmunity and inflammation of joints

To determine if the presence of anti-dsDNA autoantibodies was a general autoimmunity phenomenon, we tested another genetically resistant mouse strain, 129/SvJ. Similar to CBA/J

mice, the wild type *S*. Typhimurium and curli mutant strains displayed similar bacterial burden in the feces of 129/SvJ mice with no statistical differences measured (Fig 5A).

*In vivo* production of curli in the intestine was indicated by using the *S*. Typhimurium *csgD* luciferase reporter strain (Fig 5B). 129/SvJ mice infected with wild type but not the curli mutant developed elevated levels of anti-dsDNA antibodies (Fig 5C). When a pathologist evaluated abnormalities in the knees of the infected mice, synoviocyte proliferation was noted as a sign of inflammation [49]. Mice infected with wild type *S*. Typhimurium had significantly elevated scores for synoviocyte proliferation compared to the mice infected with the curli mutant (Fig 5D and 5E).

To independently assess the potential of curli fibers in directly stimulating autoimmunity and arthritis, we performed an additional experiment with mice subjected to weekly injections of curli purified from wild-type *S*. Typhimurium. Serum analysis revealed that these mice had elevated levels of anti-dsDNA autoantibodies with increasing doses of curli (S5A Fig). There was no statistical difference in the amount of anti-dsDNA autoantibodies developed in mice injected with either curli purified from wild type *S*. Typhimurium or curli purified from a *S*. Typhimurium *msbB* mutant, which expresses a modified LPS that doesn't signal through TLR4 ([39], S5B Fig). This result showed that the autoimmune response was driven by curli fibers and not by contaminating LPS. No significant differences were observed in serum titres of IgA, IgG and IgM upon injection of curli purified from wild type *S*. Typhimurium or curli purified from a *msbB* mutant (S5C Fig).

To determine if the presence of curli in the gut was enough to trigger autoimmunity, we performed a series of experiments. First, we orally infected 129/SvJ and CBA/J mice with wild type *S*. Typhimurium or with the non-invasive *invAspiB* mutant. Serum analysis indicated that only mice infected with the wild type *S*. Typhimurium developed anti-dsDNA autoantibodies (Fig 6A and 6B).

Next we injected mice either orally or intraperitoneally with curli fibers. Only mice injected i.p. with curli fibers developed anti-dsDNA autoantibodies (Fig 6C). Furthermore, mice injected i.p. with purified curli fibers showed significantly higher levels of synovial inflammation and synoviocyte proliferation (Fig 6D). Chronic synovitis in the joints led to periosteal bone resorption (Fig 6E, arrow), one of the hallmarks of reactive arthritis. Finally, to show that the observed autoimmunity was not restricted to curli from *S*. Typhimurium, we introduced wild type, curli-producing *E. coli* or a curli-deficient Δ*csgBA* strain by oral gavage or i.p. injections. Levels of anti-dsDNA antibodies were not increased in mice that were orally colonized with the wild type *E. coli* (Fig 6F). In contrast, mice that received wild type *E. coli* systemically had significantly elevated levels of anti-dsDNA autoantibodies (Fig 6F). Overall, these results indicated that the systemic presentation of curli fibers was required to induce autoimmunity by enteric bacteria.

## Discussion

Here, we show for the first time the presence of pathogen-produced amyloids *in vivo* in the context of an enteric infection. Curli amyloids represent the major proteinaceous component of *Salmonella* biofilms and are critical for the development of normal biofilm architecture, as they mediate cell-cell attachment and attachment to biotic and abiotic surfaces [16, 22]. To date, the optimal *in vitro* conditions described for curli expression by *S. enterica*, *E. coli* and other enteric bacteria are predicted to correspond to conditions found outside a mammalian host [14, 32]. Although anti-curli antibodies were found in human sepsis patients infected with *E. coli* [50] as well as in mice infected with *S*. Typhimurium [34], curli expression *in vivo* was doubted because the *in vitro* signals were not replicated in a living host. Nevertheless, curli

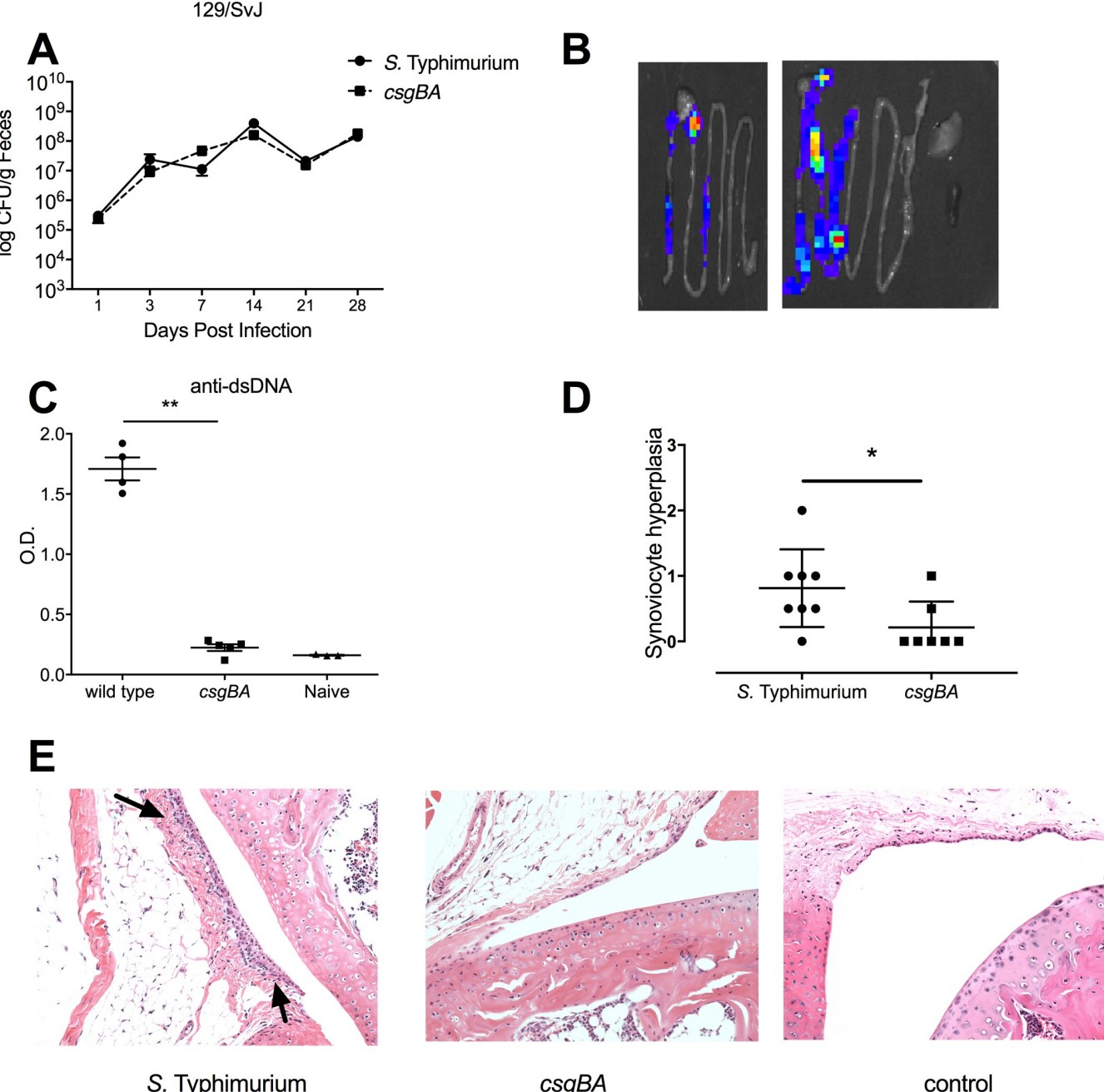

**Fig 5. Oral infection of 129SvJ mice with *S*. Typhimurium leads to the generation of anti-dsDNA autoantibodies and joint inflammation.** 129SvJ mice were orally infected with $10^8$ CFU of wild type *S*. Typhimurium or an isogenic curli mutant strain (*csgBA*). (A) The mean levels of *S*. Typhimurium CFU ± standard error was determined in fecal pellets collected on days 1, 3, 7, 14, 21, and 28 post-infection. Data shown is representative of three independent experiments. (B) At Day 4 post infection, we analyzed light production in the intestinal tract, spleen and liver from mice infected with a *S*. Typhimurium *csgD* luciferase reporter strain. (C) The mean levels of anti-dsDNA autoantibodies in mouse serum ± SE, as quantified by ELISA. Naïve mouse serum was used as a negative control. Significance was calculated using a Student's t-test (**, $p < 0.01$). (D) The level of inflammation in knee joints from *S*. Typhimurium-infected mice was scored by a pathologist, following a scale based on histological parameters: 0, no changes; 1, slight thickening of synovial cell layer (< 3 layers of synoviocytes) accompanied by congestion and edema of the external membrane; 2, moderate thickening of synovial cell layer (3–5 layers of synoviocytes) accompanied by congestion and edema of the external membrane. Significance was calculated using a Student's t-test (*, $p < 0.05$). The scoring was applied to tissue sections prepared from each infected mouse, with representative images shown in (E). Arrows denote the synoviocyte infiltration.

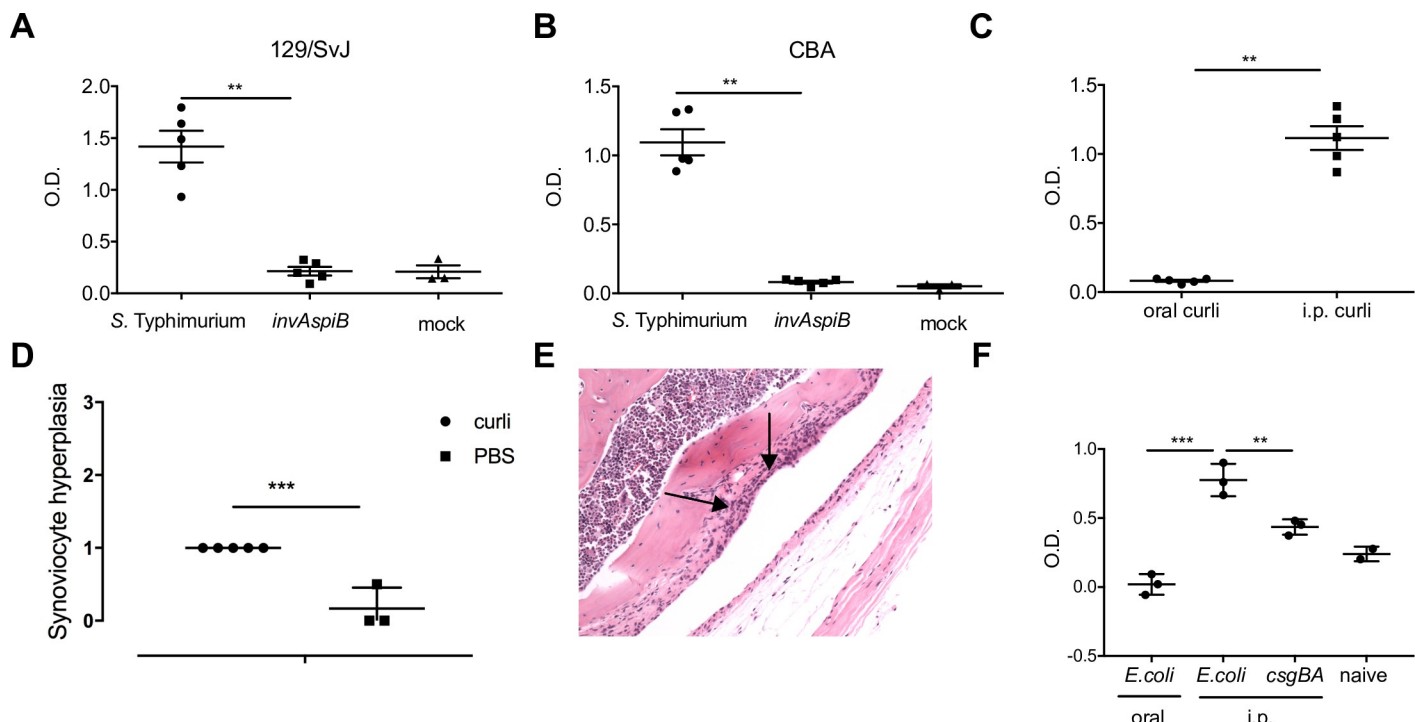

**Fig 6. Systemic curli leads to the generation of anti-dsDNA autoantibodies and joint inflammation.** 129SvJ (A) and CBAJ (B) mice were orally infected with $10^8$ CFU of wild type *S*. Typhimurium or an isogenic type three secretion–negative strain (*invAspiB*) in 0.1 ml LB Broth or mock infected with LB Broth. The anti-dsDNA autoantibodies were detected in mouse serum ± SE, as quantified by ELISA. Significance was calculated using a Student's t-test (**, $p < 0.01$). (C) C57BL/6 mice were injected orally or intraperitoneally twice weekly for 6 weeks with 100 μg curli purified from the biofilms of *S*. Typhimurium *msbB*. Mean levels of anti-dsDNA autoantibodies in mouse serum ± SE, as quantified by ELISA. (D) C57BL/6 mice were injected with 100 μg curli purified from the biofilms of *S*. Typhimurium *msbB* twice weekly or PBS for 12 weeks. Joint inflammation was scored as described above. (E) A representative image is shown with arrows denoting synoviocyte proliferation and bone resorption. (F) C57BL/6 mice were treated with 20mg streptomycin and orally infected 24 hours later with $10^8$ CFU of wild type *E. coli*. Two other groups of C57BL/6 mice were injected i.p with $10^8$ CFU wild type E. coli or curli mutant (*csgBA*) grown under biofilm inducing conditions weekly for 6 weeks. Mean levels of anti-dsDNA autoantibodies in mouse serum ± SE, as quantified by ELISA.

fibers have been established as potent inducers of the innate immune system. Curli are recognized by the TLR2/TLR1/CD14 complex as well as the NLRP3 inflammasome [38–40, 51, 52]. In addition, a *S*. Typhimurium mutant that lacks curli (*csgBA* mutant) elicits lower levels of IL-17 and IL-22 in the gut [48]. There is also evidence that curli can bind to multiple host proteins such as contact phase proteins or extracellular matrix proteins such as fibronectin and laminin [53–55]. The discovery that *S*. Typhimurium produces curli in the murine large intestine brings together years of research on differing aspects of the *Salmonella* lifecycle (i.e., life inside vs. outside the host) and reinforces the connection between persistence and virulence [11].

The signals for curli production by *S*. Typhimurium appear to be conserved within the murine large intestine. We determined that *csgD* was expressed and that *S*. Typhimurium produced curli during acute infections in susceptible mouse strains, both with and without streptomycin-pretreatment, and during chronic infections in genetically resistant mouse strains. In these models, there was no apparent colonization advantage for curli-proficient strains, which leads to questions as to why curli is produced. *Salmonella* and *E. coli* strains that are associated with invasive disease have generally lost the ability to produce curli and form curli-associated biofilms ([46, 56]; summarized in [10]), which may allow bacteria to spread systemically within the host [57]. For strains that cause gastroenteritis, studies have shown that curli and flagella are oppositely regulated by the cyclic diguanylate (c-di-GMP) signaling cascade to transition from motility to sessility [11, 58]. c-di-GMP signaling feeds into the bistable expression of

CsgD and the formation of two distinct cell types [11, 58]: multicellular aggregates, which contain curli and are adapted for persistence, and planktonic bacteria that have abundant Type III secretion proteins and are adapted for virulence [11]. The presence of two populations may help *S.* Typhimurium to better survive the hostile conditions in the intestinal lumen. Alternatively, this phenotype divergence could be an evolutionary adaptation where pre-formed biofilms help to prepare the bacteria for the external environment, which would be an advantage in the context of fecal-oral transmission [59, 60]. Such a strategy would explain why infection of mice with *S.* Typhimurium (i.e., that initially lacked curli) resulted in *csgD* expression and curli production in lower regions of the GI tract, detectable after 96 hours of infection. It should be noted that *Vibrio cholerae*, another important human enteric pathogen, undergoes similar phenotype divergence within the host GI tract [61].

We used the chronic mouse infection model to examine the immunological impacts of curli produced in the context oral infection with *S.* Typhimurium. Curli/bacterial DNA complexes are highly proinflammatory [36] and these complexes can activate multiple innate immune receptors (e.g. TLR2/1, and TLR9 [37]. Systemic exposure to curli/DNA results in the generation of type I interferons and autoantibodies against dsDNA and chromatin [37]. In murine models, intraperitoneal injection of commensal *E. coli* or pathogenic *S.* Typhimurium grown under biofilm-inducing conditions leads to elevated levels of autoantibodies as well, and these responses required the expression of curli [36]. Here we show that oral infection with *S.* Typhimurium can trigger an autoimmune response in mice within 6 weeks after oral infection, a response correlated with *in vivo* synthesis of curli. This result demonstrates that a bacterial component expressed in the intestinal tract can trigger subsequent autoimmune responses within the host. Reactive arthritis (ReA), also known as post-infectious arthritis, is a painful form of inflammatory arthritis caused after gastrointestinal infections with enteric pathogens such as *Salmonella*, *Yersinia*, or *Campylobacter* [62]. The mechanisms of how enteric pathogens induce ReA are not known. Since curli fibrils are highly conserved in enteric bacteria, it is possible that curli represent a link between gastrointestinal enteric infections and ReA cases. Histocompatibility leukocyte antigen (HLA)-B27 genotype is a risk factor for ReA, and over two-thirds of the patients with ReA carry the HLA-B27 genotype [63]. We hypothesize that the individuals that carry the HLA-B27 react stronger to bacterial curli, which may help justify the sporadic occurrence of ReA after gastrointestinal infections.

We predict that bacterial amyloids produced by enteric pathogens in the context of intestinal infection can initiate and/or exacerbate autoimmunity within the human host. In addition to *S. enterica* and *E. coli*, human commensal members of the Enterobacteriaceae can also synthesize curli [33] and curli-like amyloids are predicted to be produced by members of other major phyla inhabiting the intestine [64]. We have shown that oral exposure to purified curli alone does not lead to autoimmunity, as intestinal permeability is reduced [57, 65], suggesting that the immune system has strategies to limit the translocation of curli to the sterile tissues. However, in the context of an invasive pathogen like *S.* Typhimurium, which can cross the epithelial barrier, systemic presentation of curli to the professional immune cells leads to autoimmunity. The *in vivo* synthesis of pathogen-produced amyloids may stimulate cross-seeding interactions with other amyloidogenic proteins, such as alpha-synuclein or beta-amyloid [66–68], with the potential for long-term systemic and neurological impacts on the host.

## Materials and methods

### Ethics statement

All animal experiments were performed in AALAC-accredited animal care and use program at the Lewis Katz School of Medicine, with protocol (#4868) approved by the Temple

University Institutional Animal Care and Use Committee in accordance with guidelines set forth by the USDA and PHS Policy on Humane Care and Use of Laboratory Animals under the guidance of the Office of Laboratory Animal Welfare (OLAW). For the experiments conducted in Canada, animals were cared for and used in accordance with the Guidelines of the Canadian Council on Animal Care and the Regulations of the University of Saskatchewan Committee on Animal Care and Supply, following Animal Use Protocols #20110057 or 20170080, which were approved by the University of Saskatchewan's Animal Research Ethics Board.

## Bacterial strains and media

*Salmonella enterica* serovar Typhimurium strain ATCC 14028 was used as the wild type strain. *S.* Typhimurium strain IR715 is a fully virulent, curli-producing wild type strain derived from the 14028 strain [69]. Its isogenic mutant, IR715 *csgBA*, contains an unmarked *csgBA* deletion and was used as a negative control [48]. The SPI-1/SPI-2 type three secretion-negative IR715 *invAspiB* mutant was described previously [70]. *S.* Typhimurium expressing the luciferase plasmid under the control of the *csgD* promoter (P*csgD*::*luxCDABE*) was described previously [60]. Bacterial strains were grown in Luria Bertani (LB) broth and supplemented with either nalidixic acid (50ug/ml) or ampicillin (200ug/ml) as needed and incubated overnight at 37°C with 200rpm agitation. *S.* Typhimurium strain 14028-3b, which contains the *csgD* promoter region from *Salmonella* serovar Enteritidis 27655-3b and has constitutive production of curli fimbriae [71], was combined with the *ΔbcsA* mutation that prevents cellulose production [60]. This strain was used for curli purification. *E. coli* MC4100 and its isogenic curli mutant (csgBA) was described previously [17].

## Curli purification and generation of polyclonal immune serum

Strain *S.* Typhimurium 14028-3b *ΔbcsA* was grown overnight in Luria broth (1% salt) at 37°C with agitation and this culture was used to inoculate 120 large (150 mm x 15 mm) T agar (1% tryptone, 1.5% agar) plates using sterile swabs; bacteria were incubated at 28°C for 48 h. Bacterial cells were scraped from the agar surface using a microscope slide and the material from 20 plates was suspended in 20 mL 10 mM Tris-HCl pH 8 supplemented with 0.1 mg/mL RNAse A and 0.1 mg/mL DNAse I. Aliquots of cell slurry (1 mL) were transferred into 2.0 mL Eppendorf Safe-Lock tubes containing a 5 mm stainless steel bead (Qiagen # 69989) and were homogenized to break up the extracellular matrix using a high-speed mixer mill (MM400, Retsch; 5 min at 30 Hz), before combining into 50mL tubes. Sonication was performed to break open bacteria, with 10 x 30 s pulses at 30% amplitude using the 3mm probe and 2 min cooling on ice between pulses. $MgCl_2$ was added to a final concentration of 1 mM and the mixture was incubated at 37° for 20 min. Lysozyme was added to a final concentration of 1 mg mL$^{-1}$ and the mixture was incubated at 37°C for 40 min. SDS was added to a final concentration of 1%, along with DNase I to a final concentration of 0.1 mg mL$^{-1}$ and the mixture was incubated at 37°C overnight. The next day, cell debris was sedimented by centrifugation (25,000 x *g*, 25 min), resuspended in 10 mL of 10 mM Tris-HCl pH 8, and boiled for 10 min, and this process was repeated twice, prior to resuspension in 10 mM Tris-HCl pH 8 supplemented with 0.1 mg mL$^{-1}$ DNAse I and RNase A and 1 mg mL$^{-1}$ lysozyme; this mixture was incubated overnight at 37°C. The cell debris was sedimented by centrifugation, washed twice in 10 mM Tris-HCl pH 8, resuspended in 3 mL of SDS-PAGE sample buffer and loaded into the well of a preparative (4mm thick) SDS-PAGE gel. The gel was run continuously at 20 mA until all of the dye in the sample had run through the bottom. The insoluble material that did not enter the gel was recovered from the top of the well, washed three times in 10 mL of

distilled water, extracted twice with 5 mL ethanol, washed in 10 mL distilled water and lyophilized. This lyophilized material was resuspended in 0.2 M glycine pH 1.5, boiled for 10 min to solubilize any Type I fimbriae that were present, and the remaining debris was sedimented by centrifugation (27,500 x *g*, 25 min). The final pellet was washed five times in distilled water, resuspended in 5 mL of distilled water, transferred to a pre-weighed glass vial, frozen at -80ºC and lyophilized. At the end of this process, we had purified 300 mg of curli fimbriae. We confirmed the purity by SDS-PAGE and Coomassie blue staining, and by immunoblotting with anti-curli immune serum [32].

To generate new curli-specific immune serum, primary immunization of two eight-week old, female New Zealand White Rabbits was performed with 200 µg of curli in 500 µL PBS mixed 1:1 with diluted Complete Freund's adjuvant (diluted 1 in 10 in incomplete Freund's adjuvant to contain <0.1 mg mL$^{-1}$ Mycobacterial antigens), delivered subcutaneously in 250 µL doses at four different locations in the abdomen of each rabbit. Three booster immunizations were performed at 21-day intervals after the primary immunization, each with 200 µg curli in 500 µL PBS mixed 1:1 with incomplete Freund's adjuvant and delivered in four doses subcutaneously.

## Detection of CsgA specific IgG

Nunc Maxisorp plates (Biolegend, 423501) were coated with 1–5µg GST-CsgAR1-5 in Tris. HCL pH 8.0 and stored at 4˚C overnight. The plate was washed three times with BBS wash buffer (17.5 NaCl, 2.5g H$_3$BO$_3$, 38.1g Na.Borate in 1L H$_2$O) and then coated with 200ul per well of BBT blocking buffer (BBS + 3% BSA+ 1%Tween 20) and incubated for 1 h at room temperature with gentle rocking. After washing five times with BBS, the plate was incubated with ten-fold serial dilutions of serum samples in BBT blocking buffer and kept at room temperature for 2 h. Following washing, the plate was coated with biotinylated goat anti-mouse IgG (Jackson ImmunoRes, 115-065-071) in BBT and incubated at room temperature for 1 h. After washing, the plate was incubated with avidin-alkaline phosphate conjugate (Sigma, A7294) in BBT at room temperature for 1 h. Finally, the plate was washed five times with BBS and then incubated with 4-Nitrophenyl phosphate disodium salt hexahydrate (pNPP, Sigma-aldrich, N2765) dissolved in glycine buffer at a concentration of 1mg ml$^{-1}$ at room temperature in the dark. Optical densities were read at 650nm and 405nm using a BMG Labtech POLARstar Omega Microplate Reader. For data analysis, 605nm readings were subtracted from the 405nm readings.

## Detection of anti-dsDNA autoantibodies

ELISA was used for the detection of anti-double stranded DNA autoantibodies, following a previously described protocol [72]. Briefly, a 96-well Vinyl Plate (Corning Costar, 2595) was coated with 0.01% poly-L-lysine in 1x PBS for one hour at room temperature. After incubation, the plate was washed three times with distilled water and dried. Once dry, the plate was stored at room temperature for up to 1 week before use. The plate was coated with calf thymus DNA at a concentration of 2.5ug mL$^{-1}$ in BBS wash buffer, sealed, and stored overnight at 4˚C. The plate was washed three times with BBS and blocked with BBT for 2 h at room temperature with gentle rocking. After washing five times, the plate was incubated with a serial dilution of positive control serum (MRL-lpr or SLE), naive serum, and serum samples overnight at 4˚C. Following washing, the plate was coated with biotinylated goat anti-mouse IgG in BBT and incubated at room temperature for 1 h. After washing, the plate was incubated with streptavidin alkaline phosphate conjugate in BBT at room temperature for 1 h. Finally, the plate was washed five times with BBS and then incubated with 4-Nitrophenyl phosphate solution at

room temperature in the dark. Optical densities were read at 650nm and 405nm using a BMG Labtech POLARstar Omega Microplate Reader. For data analysis, all 605nm readings were subtracted from the 405nm readings.

## *In vivo* infection of mice

6–8 week-old female CBA/J mice, 129SvJ mice, or C57BL/6 mice were purchased from the Jackson Laboratories. The streptomycin-pretreated mouse model has been described previously [41]. Briefly, mice were inoculated intragastrically with 20mg of streptomycin (0.1ml of a 200 mg ml$^{-1}$ solution in PBS) 24 h before bacterial inoculation. Bacteria were grown shaking in LB broth at 37˚C for overnight, which results in bacteria that are devoid of curli production [9]. Mice were inoculated intragastrically with either 0.1ml of sterile LB broth (mock infection) or $10^7$–$10^8$ CFU *S*. Typhimurium 14028. Mice were euthanized at indicated time points after infection. To determine the number of viable *S*. Typhimurium, samples of cecum, liver, spleen, MLN and feces were collected from each mouse, homogenized in PBS, and 10-fold serial dilutions were plated on LB agar plates containing nalidixic acid. Organs and feces were collected for immunoblot analysis.

   *S*. Typhimurium IR715 and its isogenic *csgBA* mutant were grown with shaking in LB broth containing nalidixic acid at 37˚C overnight. These strains were used to inoculate groups of CBA/J and 129SvJ mice. Mice were tail bled and fecal samples were obtained once per week. Fecal samples were collected, homogenized in 5mL 1x PBS in 15mL conical tubes using a tabletop vortex, and 10-fold serial dilutions were performed and plated on LB agar plates containing antibiotic to detect bacterial burdens throughout the infection. Mice were euthanized six to seven weeks after initial infection. To determine the bacterial burden systemically, the liver, spleen, and fecal samples were collected from each mouse, homogenized in 1x PBS and 10-fold serial dilutions were plated on LB agar plates containing antibiotic. At point of sacrifice, blood was obtained from each mouse by cardiac puncture.

   For bioluminescence experiments, mice were inoculated orally with 2 x $10^8$ CFU of *S*. Typhimurium IR715 or its isogenic *csgBA* mutant containing the *csgD* luciferase plasmid. Bacteria were grown statically for 72h in LB low salt at 28˚C to induce biofilm formation or grown with shaking in LB salt at 37˚C overnight, which is a non-inducing condition. 129SvJ mice were euthanized 7 days after initial infection while C57BL/6 mice were euthanized 4 days after initial infection. The spleen, liver, and gastrointestinal tract were removed and imaged using the IVIS Spectrum *in vivo* imaging system (Perkin Elmer).

## Immunohistochemistry

For histological analyses, segments of the cecum, colon and liver were collected following euthanasia. Tissue segments were fixed in 10% neutral buffered formalin for 24 h prior to paraffin embedding and sectioning at 4 μm. Samples were deparaffinized and rehydrated before staining. Heat mediated antigen retrieval was performed in Tris-EDTA buffer (10mM Tris, 1mM EDTA, 0.05% Tween 20, pH 9.0) at 90˚C for 35 minutes, followed by blocking in 5% (w/v) skim milk in 1x PBS at room temperature (RT) for 3 h. Staining was performed by overnight incubation at 4˚C with staining buffer (1% BSA, 1% Donkey Serum, and 0.5% Triton X-100 in 1x PBS) containing 0.5 μg ml$^{-1}$ Goat anti-Salmonella CSA-1 (BacTrace, KPL), 1 μg ml$^{-1}$ Rat anti-ZO1 (R40.76, Santa Cruz) and a 1:100 dilution of anti-curli rabbit serum. Slides were washed before incubating with 5 μg ml$^{-1}$ each of Donkey anti-goat Alexa555 (ab150130, Abcam), Donkey anti-rat Alexa488 (ab150153, Abcam) and Donkey anti-rabbit Alexa647 (ab150075, Abcam) in staining buffer at RT for 3 h. Slides were washed, counter-stained with 1μg ml$^{-1}$ DAPI for 10 minutes, and cover slipped with VectaShield mounting media.

Imaging was carried out sequentially on a TCS SP5 scanning confocal microscope (Leica), using the 405 nm and 540 nm laser lines followed by 488 nm and 630 nm laser lines, and utilizing a 63x objective under oil immersion. Images of larger areas were generated via tiling and higher magnification achieved through digital zoom tiles using the LS ASF software (Leica).

## Curli detection in murine tissues using SDS-PAGE and immunoblotting

Two halves of the small intestine (i.e., jejunum and ileum), right lobe of the liver, cecum and colon tissues were snap frozen in liquid nitrogen and stored at -80 ºC. Tissue samples were ground into a fine powder under liquid nitrogen using a mortar and pestle. 50 mg aliquots of powdered tissues were resuspended in 500 μL of 1x SDS-PAGE sample buffer, and boiled for 10 min. The insoluble cell debris was pelleted by centrifugation (25,000 x $g$, 5 minutes), and washed twice in 500 μL of distilled water. The cell debris sample was treated with 500 μL of 90% formic acid, frozen at -80˚C for 1 h, and lyophilized for 24 h. In initial experiments, this step was performed once; in later experiments, the formic acid and lyophilization steps were repeated three times. After formic acid treatment, lyophilized samples were resuspended in 50 μL of 1x SDS-PAGE sample buffer, the samples were centrifuged (25,000 x $g$, 2 min) and a 20 μL aliquot of supernatant was loaded directly, without boiling, into each gel lane. Purified curli standards were similarly prepared, with one 20 μg sample divided between two gels. Prestained protein standards used were BluELF (FroggaBio #PM008). SDS-PAGE was performed with a 5% stacking gel and a 12% resolving gel. Proteins were transferred to nitrocellulose for 40 min at 25 V using a Trans-Blot SD semi-dry transfer cell (Bio-Rad Laboratories) in tris-glycine buffer supplemented with methanol. Monomer, dimer and higher molecular weight oligomers of CsgA were detected using rabbit anti-curli polyclonal serum (used at 1:1000 dilution), followed by IRDye 680RD goat anti-rabbit IgG (Mandel Scientific; used at 1 in 10,000 dilution) and detection using the Odyssey CLx imaging system and Image Studio 4.0 software package (Li-Cor Biosciences).

## Histopathology

Murine knees were extracted and fixed in phosphate-buffered formalin. For decalcification, samples were incubated in formic acid for 3 days and then embedded in paraffin. 5 μm sections of the tissue were stained with hematoxylin and eosin. The fixed and stained sections were blinded and evaluated by an experienced veterinary pathologist according to the criteria as described previously [49]. Images were taken at a magnification of 10x.

## Statistical analysis

Parametric test (Student $t$ test) or one-way ANOVA test was used to calculate whether differences were statistically significant ($P < 0.05$) using GraphPad Prism software.

## Supporting information

**S1 Fig. Levels of *S*. Typhimurium colonization in orally infected, streptomycin pre-treated C57BL/6 mice.** Organs were collected and bacteria enumerated from two mice from a group of six that were infected with *S*. Typhimurium 14028. (A) Arrowheads and circles represent the values from each organ for each mouse analyzed, and bars represent the mean values. (B) Two fecal pellets were collected from all six infected mice on Day 4 post-infection and the levels of S. Tm were determined. Each symbol type represents a different mouse.
(TIFF)

**S2 Fig. Aggregates of S. Typhimurium detected in infected cecum samples.** Streptomycin pretreated C57BL/6mice were orally infected with $10^8$ CFU of S. Typhimurium. Organs were collected 96 hours post-infection, fixed in 10% formalin and paraffin embedded. 5 μm sections were cut and stained with DAPI for nuclear staining or with Salmonella anti-04 LPS antibodies. FITC-conjugated anti rabbit IgG was used as secondary antibody. (A) Cecal tissue was visualized at 60X. (B) Lumenal bacteria from the cecal tissue was visualized at a higher magnification (100X oil). (TIFF)

**S3 Fig. Immunoblot screening for detection of curli fimbriae in the mouse GI tract.** Liver (LI), cecum and colon tissues were harvested from S. Typhimurium-infected C57BL/6 mice and boiled in SDS-PAGE sample buffer. The insoluble debris remaining was washed with distilled water and treated with 90% formic acid prior to SDS-PAGE and immunoblotting. Purified curli were included as a control (Pos), with arrows denoting the CsgA monomer (M) and dimer (D) species. The star represents the high molecular weight species associated with the top of the SDS-PAGE well, present in the colon sample and the positive control. Molecular mass markers (in kilodaltons) are indicated on the left. Curli bands were detected with rabbit anti-curli immune serum as primary antibody and IRDye 680RD Goat-anti-rabbit-IgG (LI-COR) as secondary antibody; immunoblots were visualized on an Odyssey CLx scanner (LI-COR). (TIFF)

**S4 Fig. Immunoblot screening for curli production in control mice.** Colon samples from 8 control mice (CM#1–8; A and B) that do not contain *Salmonella*, were processed for curli production. Samples were frozen, ground, boiled in SDS-PAGE sample buffer and the insoluble debris remaining was treated 3x successively with 90% formic acid before running the gel. Formic acid-treated, purified curli samples were included as controls (Pure curli). Rabbit anti-curli immune serum was used as the primary antibody, followed by IRDye 680RD goat anti-rabbit immunoglobulin G (IgG) secondary antibody and visualization using the Odyssey CLx imaging system and Image Studio 4.0 software package (Li-Cor Biosciences). Representative images are shown; on top, the auto-exposed images generated by the Odyssey are shown, whereas, on bottom, the same blots have been over-exposed to show that there are no faint curli bands present. (TIFF)

**S5 Fig. Anti-dsDNA immune response is specific to curli and not LPS.** Balb/C mice were immunized intraperitoneally with 10, 100 and 250 μg of purified curli (A) or with 100 μg of curli purified from wildtype *S*. Typhimurium or an isogenic *msbB* mutant (B). Injections were continued weekly for 4 weeks. At the end of 4 weeks, the levels of anti-dsDNA autoantibodies in the serum of mice were quantified by ELISA. (C) Levels of IgA, IgG and IgM in the serum of mice from (B) were quantified by ELISA. (TIFF)

## Acknowledgments

We would like to thank VIDO animal care staff for help with planning and performing the animal experiments. We would like to thank Andreas Baumler, Scott Napper, Nick Feasey, and Phil Cohen for critical reading of manuscript and Bill Kay, Marc Monestier, Roberto Caricchio, and Stefania Gallucci for helpful discussions.

## Author Contributions

**Conceptualization:** R. Paul Wilson, Aaron P. White, Çagla Tükel.

**Formal analysis:** Aaron P. White, Çagla Tükel.

**Funding acquisition:** Aaron P. White, Çagla Tükel.

**Investigation:** Amanda L. Miller, J. Alex Pasternak, Nicole J. Medeiros, Lauren K. Nicastro, Sarah A. Tursi, Elizabeth G. Hansen, Ryan Krochak, Akosiererem S. Sokaribo, Keith D. MacKenzie, Melissa B. Palmer, Dakoda J. Herman, Nikole L. Watson.

**Methodology:** Amanda L. Miller, J. Alex Pasternak, Nicole J. Medeiros, Lauren K. Nicastro, Yi Zhang, R. Paul Wilson, Aaron P. White.

**Supervision:** Yi Zhang, Heather L. Wilson, R. Paul Wilson, Aaron P. White, Çagla Tükel.

**Validation:** R. Paul Wilson.

**Visualization:** Amanda L. Miller.

**Writing – original draft:** Amanda L. Miller, Nicole J. Medeiros, Aaron P. White, Çagla Tükel.

**Writing – review & editing:** Amanda L. Miller, J. Alex Pasternak, Aaron P. White, Çagla Tükel.

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
