## [Decision Letter · Decision Letter 0]

2 Mar 2020

Dear Çagla,

Thank you very much for submitting your manuscript "In vivo synthesis of bacterial amyloid curli contributes to joint inflammation during S. Typhimurium infection" for consideration at PLOS Pathogens. As with all papers reviewed by the journal, your manuscript was reviewed by members of the editorial board and by several independent reviewers. In light of the reviews (below this email), we would like to invite the resubmission of a significantly-revised version that takes into account the reviewers' comments.

I think that Reviewer 1's suggestion to use metabolic cages will not prevent coprophagy, as mice can grab each other's fecal pellets while they are defecating (there are other ways to address this comment, though). Please pay specific attention to the major comments on experimental controls.

We cannot make any decision about publication until we have seen the revised manuscript and your response to the reviewers' comments. Your revised manuscript is also likely to be sent to reviewers for further evaluation.

Sincerely,

Leigh Knodler

Guest Editor

PLOS Pathogens

Renée Tsolis

Section Editor

PLOS Pathogens

Kasturi Haldar

Editor-in-Chief

PLOS Pathogens

orcid.org/0000-0001-5065-158X

Michael Malim

Editor-in-Chief

PLOS Pathogens

orcid.org/0000-0002-7699-2064

Reviewer's Responses to Questions

**Part I - Summary**

Reviewer #1: This study by Miller et al demonstrates that invasive Salmonella can promote arthritis “like” responses in infected mice through their expression of curli. This is explored through the use of mouse models of Salmonella infection, in vivo imaging and blotting for curli expression, as well as assessments of immune responses and tissue inflammation. Overall, the findings are largely convincing, and the implications are significant, considering the widespread interest regarding the potential for gut microbes to promote systemic autoimmune diseases. Even so, there are some additional experiments, revisions to the text and clarifications needed before the manuscript should be accepted for publication.

Reviewer #2: Amanda Miller et al

The role and importance of curli in biofilm formation and host colonization is well-documented. This is a well-written manuscript with moderately novel and at the same time thought-provoking findings. The authors claim that Salmonella produces curli during experimental mice infection and that curli is responsible for inducing autoimmunity through anti-ds DNA auto-Abs. The authors reference their own paper in which a curli-expressing Salmonella induces autoimmunity generating anti-ds DNA (ref. 36). Previous studies by Crawford R. (PNAS) showed the Salmonella produces biofilms in the gall bladder in humans, which suggest that curli is produced in the human GI tract. The present work sounds like a confirmation of previous findings. My specific comments are:

Fig.1 A Salmonella curli mutant and a complemented strain is missing in this experiment.

Line 119. I suggest to write “ 25% of the bacteria” and not “25% of cells”.

Line 124. Maybe instead of saying “individual” the sentence should read “ a very small number (how many?) of single bacteria”….

Line 185 and Fig. 4. The data in Fig. 4 suggest that curli does not promote Salmonella colonization in the gut, liver or spleen. It’s well known that curli is an important component of biofilms of Salmonella and pathogenic E. coli and that biofilm formation is key for colonization and persistence of bacteria in the gut and other host tissues and surfaces. I think the authors need to highlight this finding because it challenges the current knowledge on the role of curli. I did not see this in the discussion section either. Why is curli not important for colonization but important for biofilm formation?

Panels 4E and 5C. What does Arbitrary Units mean? Why not titers of Abs or Elisa ODs? Why was the presence of IgA Abs not tested in gut contents and sera? This is a very important question.

Line 247. It is not a natural infection. This is an experimental infection.

Line 314. What is the basis to hypothesize that individuals that carry the HLA-B27 react stronger to bacterial curli, which may help justify the occurrence of ReA after GI infections? Are there supporting data here?

Line 351, Fig. 6F. Choice of E. coli MC4100 not the most adequate one I think. MC4100 is a lab/domesticated strain that has been passed for many years in the lab. Why not a pathogenic E. coli?

Panel 6F. Please indicate inoculation routes under strains.

Immunization of animals. Nowadays, in most institutions the use of Freund’s adjuvant is not accepted. Please clarify why the use of CFA in immunizations. Do the participating institutions allow it?

Reviewer #3: The authors describe the in vivo production of curli amyloid fiber during S. Typhimurium infection in the murine gastrointestinal (GI) tract following oral infection. Using mouse models for Salmonella infection, the authors conclude that the presence of curli in the GI tract is associated with increase in autoimmune disorder as quantified by increase in the production of autoantibodies and inflammation in the infected mice. The authors further claim that the de novo curli synthesis in the GI tract of infected mice could represent a link between GI enteric infections and reactive arthritis.

**Part II – Major Issues: Key Experiments Required for Acceptance**

Reviewer #1: 1. A major issue is whether curli are expressed – de novo within the mouse intestine or if curli expression depends on the passage of Salmonella out of the gut, and is induced once they are passed into the external environment in the stool. The gene csgD is clearly transcribed within the mouse gut, but much of the luciferase signal is found in the small bowel (as shown in fig 2). During Salmonella infection, the only Salmonella that should be found in large numbers in the small bowel would be those found in stool that is eaten. Similarly, the fraction of Salmonella that express immunoreactive curli in the gut, could be those that has been ingested in the stool.

As such, the authors should assess whether Salmonella found in the stool express curli protein. Moreover, the authors should test whether limiting the ingestion of shed stool (using metabolic cages) reduces or eliminates the presence of curli +ve Salmonella in the intestines of infected mice. Even if this is the case, the study is still interesting, but it will clarify if the intestinal environment itself can induce curli expression.

2. Figures 5 and 6 - evaluation of inflammation in the joints. Considering that the majority of the readers of PLoS Pathogens will know little if anything re: joint inflammation – the data presented seems less than impressive (ie. a pathology score and some histology). I would request that the authors provide more detailed analysis – ie. immunostaning for inflammatory cells, proliferation etc, so that readers will get a better understanding of the processes involved.

Reviewer #2: The inclusion of a curli mutant and a complemented strain is important in the experiment described in Figure 1.

Anti-curli IgAs should be assayed (in addition to IgGs) in intestinal contents of infected mice. Titers of curli and ds-DNA Abs should be determined, and not "Arbitrary Units".

Experiments are needed to clarify if mice in which light was not detected (lack of csgD expression) (Fig. 2) produced or not Abs against dsDNA.

The protocol for isolation of curli from bacteria is rather complex and lengthy. However, the authors detect curli in different regions of the gut with formic acid alone. Negative controls including tissues from a mouse infected with a curli mutant and from a naive mouse should be included. Since the Enterobacteriaciae produce curli it is important to discriminate between mouse microbiota curli and the Salmonella curli.

Reviewer #3: 7. Figure 3 C western blot is missing a loading control. This panel should probably be replaced so that the data can be properly quantitated.

8. Figure 2 shows that csgD expression is detected in non-streptomycin treated animals. However, line 166-167 mentions that curli-specific bands were seen in only 2 out of 6 non-streptomycin treated mice.

9. Figure 4C immunoblots are not very clear and it is somewhat difficult to parse out the main points.

**Part III – Minor Issues: Editorial and Data Presentation Modifications**

Reviewer #1: The authors indicate that the lack of luciferase expression under the csgD promotor by Salmonella in the liver and spleen indicates that systemic Salmonella do not express the gene. This is incorrect. Luciferase detection depends on many factors, including whether sufficiently large numbers of Lux+ve bacteria are nearby each other. Thus, Salmonella may be expressing the gene csgD at systemic sites, but based on their relatively low density, they cannot be detected. As such, the authors comments should be revised.

Line 499 – spelled jejunum incorrectly ie. “Two halves of the small intestine (i.e., jejenum and ileum)”

Reviewer #2: Fig. 4C. Western blots should be improved. They are too dark and difficult to assess what is positive or negative.

Reviewer #3: 1. Line no. 35-38. The first half of the sentence is corroborated by the experiments done in this manuscript while the second half has not been addressed.

2. Supplementary Fig. S2B, scale bar is missing.

3. Figure 1E, H- The supplementary figure S1A shows colonization of the liver following oral infection of S. Typhimurium cells. However almost no bacterial cells are seen in the immunohistochemistry staining. Given the techniques, is this expected or are the particular pictures misleading?

4. Figure 2 A, B. There are four panels for each of the condition tested. Do these panels represent replicates? Please clarify in the legend.

5. Line 144-146 (Figure 2 A, B) Some of the GI tracts of the infected mice do not show any light production. In addition, there seems to be lot of variation in the expression of csgD in the GI tract of the infected animals. Could the authors comment on the variation?

6. Figure 3C. The authors collect tissue sample from infected mice and probe them for the presence of curli amyloid fibers. To depolymerize the fibers, the authors treat the tissue sample three time with 90% formic acid (FA). These three successive FA treatments are somewhat unique, as I have not seen repeated FA treatments used in other protocols. Since FA is known to fragment proteins the laddering effect seen in figure 3C western blot might be caused by the repeated FA treatments. The authors might try another denaturant.

7, 8, 9-- see above

10. Figure 4D, the data from ELISA for CsgA specific IgG production could be represented in the same format as figure 4E which shows the levels of anti-dsDNA autoantibodies in WT vs csgBA mutant infection. This would make it easier for the reader to compare the effect of presence vs absence of curli on host immune response.

11. Figure 5B- Its not clear what the two images represent. Could the relevant description be added to figure legend?

12. Line 239-240. The data for the statement is not present, but should be added.

13. Figure 6F, it’s not clear what the two “E. coli” data points represent in the figure.

PLOS authors have the option to publish the peer review history of their article (what does this mean?). If published, this will include your full peer review and any attached files.

Reviewer #1: No

Reviewer #2: No

Reviewer #3: No
---

## [Editor Report · Decision Letter 1]

27 Apr 2020

Dear Dr. Tukel,

Thank you very much for submitting your manuscript "In vivo synthesis of bacterial amyloid curli contributes to joint inflammation during S. Typhimurium infection" for consideration at PLOS Pathogens. As with all papers reviewed by the journal, your manuscript was reviewed by members of the editorial board and by several independent reviewers. The reviewers appreciated the attention to an important topic. Based on the reviews, we are likely to accept this manuscript for publication, providing that you modify the manuscript according to the review recommendations.

Dear Dr Tukel,

Thank you for submitting the revised version of your manuscript. I appreciate the efforts that you made to address the reviewer's comments by the inclusion of new data and modifications to the text. The manuscript is improved because of it. One concern remains - two of the three reviewers had asked for the addition of a negative control i.e. the curli mutant to the immunostaining of infected tissues, one of your assays used to assess the spatiotemporal nature of curli production (Figure 1). You have provided this data in your rebuttal letter and show that some mutant bacteria in the cecum do indeed react with the polyclonal antisera, which is perhaps not too surprising. Still this is a critical negative control, and I feel that quantifying the proportion of mutant bacteria by immunofluorescence that are "positive" for curli staining in the cecum provides an important comparison for the infection with wild type bacteria (of which 25% are curli-positive, line 120). I strongly urge you to add this data and resubmit the revised manuscript.

With my best wishes, Leigh

Sincerely,

Leigh Knodler

Guest Editor

PLOS Pathogens

Renée Tsolis

Section Editor

PLOS Pathogens

Kasturi Haldar

Editor-in-Chief

PLOS Pathogens

orcid.org/0000-0001-5065-158X

Michael Malim

Editor-in-Chief

PLOS Pathogens

orcid.org/0000-0002-7699-2064

Dear Dr Tukel,

Thank you for submitting the revised version of your manuscript. I appreciate the efforts that you made to address the reviewer's comments by the inclusion of new data and modifications to the text. The manuscript is improved because of it. One concern remains - two of the three reviewers had asked for the addition of a negative control i.e. the curli mutant to the immunostaining of infected tissues, one of your assays used to assess the spatiotemporal nature of curli production (Figure 1). You have provided this data in your rebuttal letter and show that some mutant bacteria in the cecum do indeed react with the polyclonal antisera, which is perhaps not too surprising. Still this is a critical negative control, and I feel that quantifying the proportion of mutant bacteria by immunofluorescence that are "positive" for curli staining in the cecum provides an important comparison for the infection with wild type bacteria (of which 25% are curli-positive, line 120). I strongly urge you to add this data and resubmit the revised manuscript.

With my best wishes, Leigh
---

## [Editor Report · Decision Letter 2]

1 May 2020

Dear Dr. Tukel,

We are pleased to inform you that your manuscript 'In vivo synthesis of bacterial amyloid curli contributes to joint inflammation during S. Typhimurium infection' has been provisionally accepted for publication in PLOS Pathogens.

Best regards,

Leigh Knodler

Guest Editor

PLOS Pathogens

Renée Tsolis

Section Editor

PLOS Pathogens

Kasturi Haldar

Editor-in-Chief

PLOS Pathogens

orcid.org/0000-0001-5065-158X

Michael Malim

Editor-in-Chief

PLOS Pathogens

orcid.org/0000-0002-7699-2064
---

## [Editor Report · Acceptance letter]

5 Jun 2020

Dear Dr. Tukel,

We are delighted to inform you that your manuscript, "*In vivo* synthesis of bacterial amyloid curli contributes to joint inflammation during *S*. Typhimurium infection," has been formally accepted for publication in PLOS Pathogens.

Best regards,

Kasturi Haldar

Editor-in-Chief

PLOS Pathogens

orcid.org/0000-0001-5065-158X

Michael Malim

Editor-in-Chief

PLOS Pathogens

orcid.org/0000-0002-7699-2064